# Protocol for evaluation of Movember's scaling what works grant funding program: Supporting the delivery of mental health interventions for men & boys in Australia, Canada and the United Kingdom

Thomas Steele[1], Chloe Ang[1], Vanessa Rose[1]*, Gayatri Kembhavi[1], Jamie Rowland[1], Maryanna Abdo[1], Janell Kwok[2], Katherine Young[1], Sophie Merryfull[3], Sara Whittaker[4], Natasha Brusco[4], Rhiannon Watt[5], Cara Büsst[5]

1 Centre for Evidence and Implementation, Melbourne, Victoria, Australia, 2 Agency for Care Effectiveness, Ministry of Health, Singapore, 3 Kinghorn Cancer Centre, Sydney, New South Wales, Australia, 4 Rehabilitation, Ageing, and Independent Living (RAIL) Research Centre, Monash University, Victoria, Australia, 5 Movember Institute of Men's Health, Melbourne, Victoria, Australia

* vanessa.rose@ceiglobal.org

## Abstract

### Background

Mental ill health among men and boys is a significant global issue, with barriers to recognising symptoms, seeking help, and accessing services. In response, Movember launched the Scaling What Works (SWW) grant funding in 2022. This initiative supports 17 diverse mental health projects across Australia, Canada, and the UK, targeting communities, schools, and workplaces to address the varied needs and contexts of men and boys.

### Methods

Our evaluation protocol outlines the approach to assessing the SWW program across four domains: implementation, effectiveness, cost (implementation and cost-effectiveness), and scalability. Using the Intervention Scalability Assessment Tool (ISAT) as a framework, we aim to embed scalability considerations into the evaluation design. Data will be collected qualitatively and quantitatively from project participants, facilitators, and Movember staff throughout the funding period. Effectiveness will be measured using the Personal Wellbeing Index (PWI) as a universal outcome across projects. Scalability will be assessed using a purpose-built tool developed in consultation with ISAT's creator, tailored to project-specific needs.

**Data availability statement:** No datasets were generated or analysed during the current study. All relevant data from this study will be made available upon study completion.

**Funding:** Movember has commissioned the research team led by the Centre for Evidence and Implementation to conduct this study, to the total funded amount of $1,582,757 AUD (authors: TS, CA, VR, GK, JR, MA, JK, KY, SM, SW, NB). The study has been funded by Movember as part of the Scaling What Works Program (https://au.movember.com/story/scaling-what-works-grants-awarded; https://au.movember.com/). There is no award associated with the funding of this study. No authors received salary or other funding from commercial companies. Staff from Movember (RW, CB) have been involved in the review of the study design, decision to publish and the preparation of this manuscript. The funders had no role in study design, data collection and analysis, decision to publish, or preparation of the manuscript.

**Competing interests:** The authors have declared that no competing interests exist.

## Conclusion

This evaluation will provide insights into program effectiveness, implementation strategies, delivery costs, and cost-effectiveness. With a strong focus on scalability, it aims to inform mental health service providers and funders on best practices for scaling interventions within grant-funded contexts.

## Introduction

The burden of men's mental ill health is significant, manifesting in poor health system access and use, disproportionate rates of suicide [1–3] and an economic impact estimated at around USD 5 trillion globally [4]. Men and boys experience unique, gender-specific barriers to mental health help-seeking related to stigma, pressures to conform to traditional masculine norms (e.g., self-reliance, stoicism), and fear of social repercussion for non-conforming behaviours [5–9]. Moreover, men are less likely to recognise and act on symptoms of poor mental health than women [7,10]).

This brings into focus the urgent need for *effective interventions/programs* that are *scalable*, those well-placed to improve access for more men and boys. Within these programs, there is also the need to develop and test gender-responsive mental health approaches to account for the differential needs, concerns, preferences, identities, and contexts of men and boys [11]. The delivery setting of men's mental health promotion programs are considered a key part of this solution (e.g., in community centres, schools, or workplaces) [12,13]. Studies have found that programs integrating gender-sensitive or gender-transformative approaches in familiar environments (e.g., within existing communities or in an all-male environment) can provide a safe space to build trust and confidentiality and encourage men to 'open up', resulting in men accessing needed healthcare [7,9,12,14].

In addition to the learnings of what makes mental health programs effective for men in all their diversities, understanding of how these are implemented remains a priority area for dedicated research. Two recent systematic reviews have both concluded that mental health programs targeting men and boys require further development, refinement, and evaluation to meaningfully understand their *impact* [13,15]. Implementation insights in this context are vital for achieving desired mental health outcomes at an individual or group level, but are also applicable to a community, system or potentially population level, through informing how social and environmental determinants of health can be incorporated into best-practice mental health policy, programming and approaches.

Furthermore, the scaling of effective, easily- and well-implemented programs to reach and improve the mental health of more men and boys is an area of sharp focus across the sector. Scaling is an emerging field of scientific research defined by the World Health Organisation [16] as the "deliberate effort to increase the impact of successfully tested health interventions so as to benefit more people and to foster policy and program development on a lasting basis." While the origin of the scaling literature lies predominantly in clinical healthcare interventions, scaling is highly relevant

for community mental health programs targeting men and boys. However, scaling within the context of social programs brings its own complexities and challenges which are often greater than other ventures [17]. For example, scaling is often conflated with the subtly different concepts of replication and expansion [18], which for social programs can become tantamount to organisational growth and operation at a larger scale. This is especially true in organisations for which program delivery is existential for business and commercial viability and is further underscored for programs and organisations actively seeking to reach underserved groups and to fill gaps in the social service system [19]. The unique challenges faced by social programs and enterprises regarding scaling emphasises the importance of understanding 'scalability', or the *extent* to which a program *can* and *should* be scaled.

While numerous frameworks and tools exist to assess and understand the 'scalability' of programs, the evidence base is limited by the tools' incongruousness, their heretofore narrow breadth of application, and the paucity of pragmatic evaluations testing them [20,21]. Moreover, the motivation for assessing the scalability of interventions appears, among published studies, to serve one or both of two purposes: 1) to inform the refinement of an intervention (e.g., to better fit the delivery context so that the intervention becomes more scalable); or 2) to establish whether scaling is possible for this intervention (in itself, or compared to an alternative) [18]. Zamboni and colleagues (2019) describe these as *formative* and *predictive* purposes, with acknowledgement that the purpose of scalability assessments is seldom reported, let alone articulated in these terms. These limitations, together with pragmatic, real-world pressures to scale interventions [22] pose a challenge to the collection of transferrable insights. This in turn poses challenges in advancing the scaling knowledge base and progressing understanding about scaling to effective programs to successfully broaden coverage to a population in need, in this case, mental health and suicide prevention programs for men and boys.

## Scaling what works program – Movember

Movember is a global men's health charity that seeks to address major health issues faced by men and boys. Since its inception in 2003, Movember has supported more than 1,320 men's health projects globally, including numerous research studies, to help transform the way health services reach and support men [23]. The Movember Institute of Men's Health is an international innovation and learning hub dedicated to building capacity in men's health research and services. In 2022, Movember launched the AUD$10.38 million Scaling What Works (SWW) grant funding program to support the scale-up of 17 promising community-, school- and workplace-based initiatives focused on improving the mental health and wellbeing of men and boys.

The multi-year SWW program includes an evaluation, providing an opportunity to contribute to the evidence base in three ways. First, the evaluation can build on the current understanding of effective approaches to men and boys' mental health promotion. Second, it can contribute to the scaling knowledge base with a particular focus on men's and boys' mental health programs. Finally, it serves as a novel example of a methodological approach for evaluating diverse interventions at a whole-of-fund-level, particularly in relation to scaling, which could provide a blueprint for similar grant-style initiatives and philanthropic organisations working in this area.

## Evaluation aims and questions

The aims of this evaluation are to assess the implementation, effectiveness, cost of implementation, and cost-effectiveness of the SWW funding program and to develop a deeper understanding of the factors that influence the scaling and sustainability of men's and boys' mental health programs (hereafter known as "projects" to differentiate between the individual funded interventions and the whole SWW funding program (the "SWW program")). The five evaluation questions are:

1. To what extent have projects been implemented as intended?

2. To what extent have participant outcomes changed over time (within projects and across the SWW program)?

3. What are the costs to deliver the intervention per participant, per individual project, and across the SWW program?

4. What is the cost-effectiveness of projects across the SWW program?

5. To what extent are projects sustainable and ready to scale further?

## Methods/design

A pragmatic approach was adopted to evaluation of the SWW program, at both an individual project and SWW program level, making the best use of available data sources, whilst aiming to minimise the disruption that might be caused by primary data collection for funded project delivery teams. Participant recruitment is still being undertaken, which is estimated to be completed in December 2025. Data collection is estimated to be completed by February 2026, with results expected by July 2026. The evaluation was approved by the Bellberry Human Research Ethics Committee (2023-01-002-A-2) and received governance approval from Monash University Human Research Ethics Committee (Project Number: 45709). The economic evaluation will be reported in accordance with the CHEERs checklist [24].

## Evaluation design

The study will utilise mixed-methods design, selected in response to the evaluation scope and breadth of questions – focusing on effectiveness, cost, implementation and scalability. The ISAT (Intervention Scalability Assessment Tool) [25] was used as an initial organising framework for the evaluation with a view to ensure that scalability would be embedded in the design. Responses to the five evaluation questions will be informed by data collected from individual projects which will be analysed at the project and SWW program level. S1 File provides a map of evaluation questions with corresponding data collection methods/information sources. Quantitative data will be used to evaluate project outcomes, cost of implementation, and cost effectiveness. Qualitative data will be used to evaluate project implementation and scalability. Mixed methods will be used to evaluate factors influencing implementation costs. These will be collected from the perspectives of both the funded project teams and the funder (Movember). Using Palinkas et al.'s taxonomy [26,27], quantitative and qualitative data will be collected and analysed simultaneously, and datasets will be merged, where appropriate, to complement and elaborate findings across the funded projects and/or evaluation components.

## Evaluation participants

Movember selected 17 diverse projects located in Australia, Canada, and the United Kingdom for funding under the SWW program, targeting boys and men aged 12 years and above in the workplace, school, or community settings (snapshot in Table 1 below, for more detailed descriptions of the interventions, see S2 File). Submissions from the Republic of Ireland were eligible for SWW, though no applications were successful in being included within the SWW program. Projects selected for funding were heterogeneous: each targeted different ages and backgrounds of boys and men and were varied in their level of establishment: both in terms of the evidence underpinning them (i.e., the quality and quantity of evidence to support the project- which ranged from published effectiveness trials to interventions at a pilot stage) and the extent to which projects are being implemented (i.e., the number of sites/cohorts implemented prior to applying for the SWW program). Funded organisations include universities and community not-for-profit organisations. Under the conditions of the SWW program, each organisation defined boys and men 'at-risk' for poor outcomes in a way that reflected the context of their individual target populations and projects.

There are three types of participants in the evaluation: 1) Project teams funded by Movember under the SWW program. These teams will nominate staff directly involved in the management and/or delivery of projects to participate in data collection of implementation, scalability, and cost data of individual projects. These staff members should be able to read and write English; 2) Participants of the 17 SWW projects who meet the following inclusion criteria:

**Table 1. Snapshot of the 17 projects funded by the SWW program.**

| Project name | Country | Target group | Delivery setting |
|---|---|---|---|
| Dads Tuning into Kids | Australia | Fathers of 3- to 12-year-olds who are already participating in fathering groups. | Community |
| South-Western & Western Sydney Men's Mental Health & Gambling Harm Prevention Project | Australia | Men aged 16 and above from Western and South-Western Sydney who are experiencing (or at-risk of experiencing) gambling harm and associated mental health challenges. | Community |
| Western Bulldogs Sons of the West Men's Health Program: Engaging with CALD communities | Australia | Culturally and linguistically diverse (CALD) men (e.g., South Asian, Vietnamese, and multiple groups within the African-Australian diaspora) of all ages living in the West of Melbourne. | Community |
| Top Blokes Mentoring Program | Australia | Male students aged 10–17 residing on the Sunshine Coast who are at risk of, or currently experiencing, mental health challenges, social disconnection, or have a diagnosed emotional or behavioural condition. | Schools |
| Our Futures | Australia | Young men aged 13–17 in participating schools and youth services across Australia. | Schools |
| Preventure | Australia | Young men aged 13–15 in participating schools and youth services across Australia who are identified as high-risk (high scores on measures of at least one of four personality traits associated with an increased risk of substance use and mental health problems). | Schools |
| Edge of the Present | Australia | Young men under 25 who are socially disadvantaged and/or living in regional and remote communities who face a wait for services. | Community |
| Scaling WiseGuyz to Youth Criminal Justice Settings | Canada | Young men aged 12–17 years who are in conflict with the law and are involved at John Howard Society sites, or other partner sites, in Canada. | Youth justice |
| RISE YBMen Toronto | Canada | African, Caribbean, and Black young men aged 16–30 who have experienced the homicide of a family member or friend. | Community |
| Collective Resilience by Working with Men in Sports and Community Settings | United Kingdom | Men living in the UK, with priority given to individuals from socially disadvantaged backgrounds. | Community |
| They Call Me Dad | United Kingdom | GBTQ+ parents and prospective parents who are experiencing challenges associated with the transition to parenthood. | Community |
| Achieving Active Lives | United Kingdom | Men experiencing early issues with their mental health, with a particular focus on communities experiencing social disadvantage and unemployment. | Community |
| Sport in Mind | United Kingdom | Men aged 16 and above who are struggling with their mental health and are not engaging in traditional mental health services. | Community |
| Growing2gether | United Kingdom | At-risk young men aged 13–16 years in Scottish Highland areas, Dundee/Aberdeen who are in remote and/or deprived areas. | Schools |
| Becoming a Man | United Kingdom | Young men aged 12–16 years from six schools based in London boroughs Lambeth and Islington who are facing challenges with their social and emotional development. | Schools |
| Offload | United Kingdom | Men aged 16 and above who are identified to be at risk of mental ill health and are living in Northern England and working in the construction industry. | Workplace |
| Good Vibrations | United Kingdom | Men aged 50 and above in Northern Ireland. | Community |

- identify as male,

- aged 12 years and above, and

- enrol in and attend at least one project session (as defined by each funded project), and;

3) A nominated Movember representative. To understand the true costs of implementation as part of the economic evaluation, a representative from the funder (Movember) with visibility of the SWW Program will participate in data collection for this component only.

## Data sources and data collection

### Project teams.

1. *Administrative and program data.* Project delivery and administrative data, including project participants' demographic and engagement data (e.g., number of project sessions attended), will be collected by project teams throughout the delivery of their projects as part of ongoing monitoring and submitted to the evaluation team every six months. Project teams additionally send quarterly project status reports to Movember with de-identified, aggregated engagement data as part of their funding agreement.

2. *Implementation focus group.* We will invite project team members to participate in a fit-for-purpose implementation focus group discussion led by at least one member of the evaluation team during the final months or shortly after completion of the funded program's delivery period. Multiple team members will be invited, to enable the evaluation to gather diverse perspectives from across each team (e.g., facilitators, project managers, project leads, analysts). An explanatory statement will be provided before informed, written consent in the form of a signature is obtained from participating project team members. The implementation focus group will follow a semi-structured interview guide which covers project delivery and fidelity, project acceptability (including understanding the application of gender-specific approaches), the barriers and facilitators to implementing their project, any strategies used before or during implementation (as per Powell et al. 2015 [28]), and project team members' perceptions of the implementation of their project. Interviews are expected to be completed in 50–60 minutes. Discussions will be audio recorded and transcribed for qualitative thematic analysis.

3. *Scalability Assessment.* Movember approaches scaling in a collaborative and supportive way with front-line delivery organisations, and the fund encompasses diverse organisations and intervention types, settings, and populations by design. The evaluation will therefore assess the scalability of SWW projects using a tool developed specifically for this evaluation and program context (evaluating heterogeneous projects funded under one funding program). The tool was informed by the ISAT [25] and findings of a literature review of existing scalability tools and approaches [18–21,29–33] which identified a knowledge gap in the field in terms of scalability in a grant making/funder and recipient dynamic, and how best to support one another given these perspectives and experiences. The developer of the ISAT, Professor Andrew Milat was consulted during this process and contributed toward the development of the tool. The tool will be partially pre-populated by the evaluation team as part of the assessment with relevant available information (e.g., program logic and implementation plans provided earlier in the evaluation) and will be administered in an interview format by the evaluation team during the final months or shortly after completion of each program's funded delivery period. This involves key SWW project team member/s (e.g., project leads, lead facilitators) being asked questions pertaining to 25 items within seven domains of the assessment tool ('The project/intervention', 'Scaling vision', 'Evidence for scale', 'Marketability', 'Delivery organisation', 'Local fit', and 'System fit'). Each item is rated according to one of three predefined categories: 'Considered', 'Progressed' and 'Established'. The assessment tool takes approximately 30 minutes of pre-population and a discussion period of 45–60 minutes to complete. Discussions will be audio recorded and transcribed for qualitative thematic analysis. As we foresee that the scalability assessment and implementation focus group discussions will be likely to occur within a single sitting (due to capacity and preferred scheduling of project team members), the approach to obtaining consent within the implementation focus group provides information about and is therefore inclusive of the scalability discussion. If an individual is to participate in the scalability discussion only (i.e., not in the implementation focus group), consent will be obtained in writing via email and/or verbally with the participant/s prior to the discussion. Throughout the evaluation, the tool will be piloted and iterated based on feedback from evaluators and project teams and via consultation with Movember and scaling experts regarding its utility.

4. *Cost data.* A costings survey (fit-for-purpose online survey) will be completed by nominated project team members for each of the SWW funded projects, and once by a representative of Movember, using the survey platform REDCap

(Research Electronic Data Capture) [34]. REDCap electronic data capture tools are hosted at Monash University and managed by Helix [35]. REDCap is a secure, web-based application designed to support data capture for research studies, providing 1) an intuitive interface for validated data entry; 2) audit trails for tracking data manipulation and export procedures; 3) automated export procedures for seamless data downloads to common statistical packages; and 4) procedures for importing data from external sources. The survey has been pilot tested for user legibility, clarity and efficiency. The consent procedure for the survey is built into the landing page of the survey, whereby an explanatory statement is presented for the project team to read and check a box to state their consent to participate. The costing survey will collect data on costs related to implementation, including: fixed costs, which do not change with the number of participants (e.g., rent, venue hire, utilities, overheads and insurance); stepped fixed costs, which may change with service volume (e.g., staff Full Time Equivalent (FTE) for administration (e.g., front desk/reception), equipment, technology, advertising, supplies and materials); variable costs, which are considered as costs that may change with the number of participants in the program, (e.g., staff FTE who provides intervention, staff training costs, travel, and refreshments); in-kind costs; and factors that may have influenced the costs to implement the program. Research related costs as part of the evaluation will be collected separately to the intervention costs as part of this survey. The costing survey will take approximately 30–45 minutes to complete. One costing survey will be completed per project, and project team members involved in resource and finance management will be encouraged to complete the survey together to ensure the data is comprehensive. Planned costs for individual programs will be provided to the evaluation team as per the funder budget. The costing survey will be completed by a nominated representative from Movember to understand program-level associated costs such as program management.

5. *Informal qualitative data collected during ongoing engagement with project teams:* The evaluation team will have ongoing engagement with the project teams throughout the course of the evaluation. Where appropriate, insights and observations about project implementation and scalability from the project teams will be collected and summarised opportunistically, via regular, scheduled interactions with the project teams.. These sources include meeting notes, project logs and updates provided by members of the evaluation team. As this information is specific to the evaluation, the requirement for consent was waived by the ethics committee.

**Project participants.** Participants (men and boys) from all SWW-funded projects who meet the eligibility criteria will be invited to participate in the evaluation by completing surveys at baseline and follow-up as part of the evaluation. During each survey, participants will be asked to complete the Personal Wellbeing Index (PWI) [36]. The PWI is a psychometrically validated tool that measures subjective wellbeing through satisfaction across seven life domains with age-adaptations for adults and adolescents [37]. PWI was thus selected as a common outcome across the SWW program evaluation, as all SWW projects have a shared aim of improving the wellbeing of its participants. The 7-item PWI is low-burden in terms of brevity and simplicity, has been psychometrically tested for construct and convergent validity, reliability and sensitivity, and is available in different age and language versions.

In addition to the PWI, each project team can nominate up to two additional outcome measures (hereafter referred to as "project specific outcomes"), which align to the project's delivery setting, mechanism of action and intended wellbeing outcome(s). These measures will be included in the SWW evaluation and analysed at the individual project level.

*Participant baseline survey:* The baseline survey includes demographic questions, the PWI, and the additional project-nominated measures.

*Participant follow-up survey*: The follow-up survey includes the same measures collected at baseline and includes both closed- and open-ended questions on participants' perception of project accessibility, acceptability, and effectiveness.

An adult and an adolescent version of the participant surveys will be created using the appropriate versions of the PWI and will be made available to the projects as required based on their target populations.

Project teams will be responsible for collecting survey data from eligible project participants. Data collection processes are designed to be flexible in terms of accommodating the variability across project design, approaches and delivery

settings. Project teams have prepared detailed implementation plans for the evaluation team and funder to describe their intervention in more detail and support embedding the evaluation within it. The evaluation team will provide relevant materials (e.g., information sheets, consent forms, data collection templates), train the project teams on the data collection process, and conduct data quality checks once data is being collected. Project participants who consent to participating in the evaluation will be asked to complete the baseline survey at project intake or within three weeks of starting the project, if the former was not feasible to collect. The follow-up survey will be administered upon project completion or exit by project teams/facilitators. For projects with set duration and/or specified number of sessions, this will occur at the final session. Projects which do not have a specified start and end date (i.e., operate on a drop-in basis) will administer follow-up surveys three months after the date of the participants' first attended session. Participant surveys can be completed online and/or using paper-based forms, whichever is deemed most appropriate and feasible for the project teams and their participants.

**Consent procedures.** Written consent will be obtained from project participants via signing a consent form attached to an information sheet. This process is administered by project teams (i.e., project facilitators/staff). The evaluation will provide project teams with information sheets and consent forms and training in these procedures and requirements.

For projects targeting adolescents below the age of 18, information sheets and consent forms will be provided to both the adolescent and their parents/guardians. Written, informed consent from both the adolescent and their respective parent/guardian is required for the adolescent to participate in the evaluation study. Adolescents who have the status of an independent minority will not require parental/guardian consent.

As it is anticipated that funded projects within the SWW program will vary between in-person and virtual delivery (e.g., online video conferencing and collaboration software), these factors are accounted for in the approach to obtaining consent:

- Projects delivered in-person: Project teams will be briefed by CEI on how to obtain informed consent as outlined above (e.g., detailing the information sheet with potential participants, not using coercive language, clarifying that participation in the evaluation is entirely voluntary, directing participants and/or their parents/guardians to the evaluation team if further questions arise, and giving participants sufficient time to make a decision). Project teams will administer information sheets and consent forms on paper and obtain written consent from SWW project participants.

- Projects delivered virtually: Project teams will be given a link to an electronic information sheet and consent form on Qualtrics set up by the evaluation team. Project teams will send out the link to participants to obtain consent electronically prior to survey completion.

## Analytical approach

**Quantitative data.** The quantitative data to be analysed includes project administrative and participant survey data collected using closed-ended questions. This does not include the quantitative economic data, which is described in the subsequent section below. Table 2 outlines the analysis approach for the different types of quantitative data. Evaluation reports will provide descriptions of quantitative data at baseline and follow-up.

Table 2. Analysis approach of quantitative data.

| Data source | Frequency of collection | Analysis approach |
|---|---|---|
| Administrative (project participation) data (attendance, completion, dosage). | Collected throughout each project's delivery | The data will be analysed using descriptive statistical analysis. |
| Project participant (pre-post survey) data: PWI, project specific outcomes. | Collected at two points in time (pre- and post-project participation) | The data will be analysed using mixed-effects models. |
| Project participant (post-survey) data: Project accessibility, acceptability, appropriateness, and feasibility. | Collected at the follow-up (second) time point. | The data will be analysed using descriptive thematic and where appropriate, statistical analysis. |

Data will be analysed using R Studio, Version 4.4.2 [38] and SPSS Statistics Software, Version 24 [39]. The specific models to be used will be identified once the characteristics of the data are more apparent. Broadly, we anticipate conducting repeated-measures analyses of covariance (ANCOVAs), with covariates for participant characteristics (e.g., participant age, attendance etc.) as relevant to each project. Program level analyses will additionally include a covariate for each project. This will allow evaluation of the impact of the projects on wellbeing outcomes. Where relevant, post-hoc tests will also be conducted to explore specific patterns of effects observed. The analysis will be complemented by visual representations of the data.

**Economic data.** *Cost of implementation*: Cost of implementation will be established for each project within the SWW program. This will include real and in-kind resources and activities that will be allocated to different cost categories (fixed costs, stepped fixed costs and variable costs). Differences between the actual costs to implement the program versus the planned (budget) costs will be collected for comparison. For each project, total cost and cost per participant will be reported. Across the projects, the combined cost, as well as the average cost per project, and per participant will be reported (mean and standard deviation). In addition to the cost of implementing the program, the cost of evaluating the program will be reported (i.e., research-related activities). These costs will be reported separately and combined with the cost of implementing the intervention. For the cost of implementation data, modelling will not be performed due to the analysis being straightforward in calculations.

Qualitative data will be gathered as part of the costing survey describing the factors, if any, associated with the increases or decreases in costs. This will include an explanation from the project team of how these factors impacted costs. Any other open text responses in the costing survey will be analysed as qualitative data. Two researchers will code the qualitative data independently using NVivo software [40]. Through a consensus process after coding with two researchers, concepts will be grouped into sub-themes and these will subsequently form main themes. The economic evaluation will not use a predefined framework and instead will be derived from the data and the number of concepts, allowing for as many concepts to be identified as they emerge. A third researcher will be engaged should consensus not be reached on the coding framework. The coding framework, and themes supporting the interpretations of the thematic analysis will be reviewed by the evaluation team at CEI, and Monash University researchers.

*Cost-effectiveness and effect analysis*: Economic efficiency will be expressed in terms of incremental cost per one-point improvement on the PWI. The cost-effectiveness analysis will include participants of SWW programs that have consented to the study. For the cost outcome, implementation cost per participant will be included, excluding research costs. For the effect outcome, the mean difference of scores for PWI from baseline to end of program delivery (or at three months post-baseline) will be used. For each project, the incremental cost per one-point improvement on the PWI will be established and compared for economic efficiency. Programs will be categorised based on program characteristics, such as age group of participants, and individual versus group interventions. Programs will be categorised considering program characteristics, such as age group of participants, and individual versus group interventions.

Cost data will be presented in Australian dollars (AUD). Raw data provided in currency other than in AUD will be converted to AUD. Once in AUD, all costs will be inflated accordingly by consumer price index (CPI) and presented as a consistent base year (e.g., 2024/2025 financial year). The economic evaluation assumes a default funded delivery period of 2 years (i.e., from the start of the project funding period to the end of the project funding period). Individual projects may be granted extensions by the project funder, and these will be reported as part of the results. Discount rates will not apply as costs will be inflated to the last financial year of data collection (e.g., 2024/2025). Data will be analysed using SPSS Statistics Software, Version 24 [39].

**Qualitative data.** *Participant survey data*: The qualitative data to be analysed includes the participant survey data collected through open-ended questions pertaining to project acceptability, appropriateness, and feasibility (follow-up survey only) and the data collected as part of ongoing engagement activities between project teams and the evaluation team (e.g., meeting notes). The qualitative survey data will be categorised and summarised using content analysis using indicative, de-identified quotes where appropriate.

 

*Project team focus group data:* Focus group audio recordings will be transcribed and analysed thematically according to main components of the semi-structured interview [41]. For example, themes will include implementation barriers and facilitators, implementation strategies used and perceptions of project acceptability, appropriateness, and feasibility among those responsible for its delivery. Analysis will occur in two stages: first at the individual project level, coded by at least one evaluator, and second, at the whole SWW program level to understand the prominent themes across all funded projects. Each analysis level will still be reported on as part of the final evaluation reporting structure, meaning there will be individual summaries for each project as well as the evaluation report at the whole of fund level.

*Project team scalability assessment data:* The scalability assessment tool itself will be analysed as a list of results for each item, classified as either: 'Considered', 'Progressed' or 'Established'. Qualitative data collected via audio recordings of the scalability conversations will be transcribed and analysed thematically according to the domains of the scalability tool. This will provide additional insight about the perceived relevance of each domain to project teams as they approach scaling. Analysis will occur in two stages: first at the individual project level, coded by at least one evaluator, and second, at the whole SWW program level to understand the prominent themes across all funded projects. Each analysis level will still be reported on as part of the final evaluation reporting structure, meaning there will be individual summaries for each project as well as the evaluation report at the whole of fund level.

*Informal qualitative data collected during ongoing engagement with project teams:* Data collected as part of ongoing engagement activities with the project teams will be summarised and categorised into themes (for example, as implementation challenges documented by project teams) which can be mapped to the EPIS (Exploration, Preparation, Implementation, Sustainment) model [30] as an organising framework to contextualise project experiences throughout the funded delivery period.

## Discussion

There are several features of this protocol which made it challenging to design and to demonstrate significance to the field. First, this protocol was designed for the evaluation of a complex grant funding program comprising 17 individual projects across three countries. This necessitated a pragmatic evaluation approach comprising four components: effectiveness, implementation, economic, and scalability. Second, the individual mental health projects for men and boys funded under SWW made the design challenging due to their diversity (i.e., differences in delivery mode, delivery setting, target population, mechanism of change, location and project teams, and delivery timelines). It was necessary to select a common outcome, the PWI, that was sufficiently broad to encompass all projects and minimal burden in terms of data collection. We acknowledge that it is possible that the PWI may not always be sensitive to the diverse range of changes experienced by project participants. We anticipate that the breadth and scope of this evaluation design would render it applicable to other grant funding programs or initiatives involving multiple disparate interventions within them.

One unique feature of the evaluation design is that scaling and scalability underpins the overarching approach to information gathering and evaluation, such as:

• The information gathered and presented to provide an overview of the SWW program and its projects, and to inform the design of the evaluation plan, was informed by the ISAT.

• The approach to assessing scalability as part of the evaluation design has allowed for the development of a tool which, inspired by ISAT, has been developed with the specific objective of addressing a gap in the field: practical insights of scaling mental health projects from grant maker and funding recipients together within an evaluation. We anticipate that the insights generated in this evaluation will benefit philanthropic organisations, governments, service delivery organisations and researchers/evaluators alike.

• The assessment of scalability for the interventions included in the SWW program will consider scalability according to the proposed domains, using both pre-existing information on the interventions being scaled (e.g., project

documentation and prior evaluations), the evaluation findings on how the scale-up was delivered, and the political, strategic and cultural context within which the intervention exists.

## Strengths

This study proposes four tiers of evaluation of the SWW program, comprising implementation, effectiveness, economic, and scalability components. The evaluation will provide insights at both a project and fund level. Evaluation at the fund level in this context is unique as it presents opportunities to understand implementation successes and challenges faced by a range of different organisations concurrently, and a platform with which they can share experiences not only with the funder, but also one another. Movember has recognised this, through implementing a dedicated 'Knowledge Exchange' community of practice aspect within the SWW program. The diverse nature of the individual projects supported by the SWW program presents an opportunity to understand the nuances between projects that are responding to different needs, including different approaches to incorporating considerations of masculinity into intervention design. The evaluation approach enables project-level insights to be surfaced, gathered, and presented both individually as well as collectively across the program, offering unique insights and learnings that can potentially be applied in other contexts. Moreover, administrative data collected from project staff will provide insights into the implementation of individual projects and qualitative data will complement these by portraying the experience of implementation – by understanding barriers and facilitators. Project-level evaluation will also contribute to insights about scaling and scalability. These insights will be shared across projects to build the knowledge base about community-based mental health interventions for men and boys.

A strength of this evaluation is the approach to assessing program and project effectiveness simultaneously. In contrast to research investigating outcomes of a single intervention, this evaluation incorporates a common measure, the PWI, as a universal method to estimate the collective effect of projects funded by the SWW program on the wellbeing of participants, while also collecting data on project specific outcomes. Such findings from fund-level evaluations can drive and inform effective decision-making among program developers and funders in mental health in men and boys. This evaluation also provides an opportunity to build understanding that could contribute to the intervention design, and implementation and scaling strategies relevant to mental health promotion for men and boys. The economic evaluation included in this study provides a comprehensive analysis of the costs to implement the program per participant, per project, and across all projects within the SWW program, and the opportunity to demonstrate economic efficiency of the SWW program by expressing the cost per one-point improvement on the PWI guiding future scaling of individual programs to the program funder (Movember).

## Limitations

Conclusions from contemporary research on the effectiveness of men's mental health interventions are difficult to draw due to the heterogeneity of programs and participants. Similarly, the projects funded by the SWW program cover a range of projects that vary by intervention focus and target population. This may limit the comparability across projects, particularly in relation to participant outcomes.

The PWI was selected as a common outcome measure as all SWW projects aimed to improve the wellbeing of their participants. However, given the heterogeneity of projects, items within the PWI may be more contextually relevant to some projects than others. Moreover, while the PWI was selected to reduce data collection burden, it does not allow for more diverse measures of wellbeing. While the evaluation will collect data for up to two additional, project specific outcomes, two key limitations arise: 1) the tools nominated by teams to collect this information have varied degrees of psychometric assessment, 2) the differing wellbeing focus of individual projects will only be reportable in isolation and cannot be meaningfully compared with other funded projects. From an economic evaluation perspective, quality adjusted life years (QALYs) were unable to be determined as the PWI was unable to be converted to a utility index.

Several data collection challenges have also emerged. First, the project teams have varying experience and different constraints with data collection and evaluation activities. To overcome this challenge, the evaluation team has been providing support to project teams to ensure that data is collected and recorded appropriately. Second, collecting data from participants over two timepoints may be more challenging for flexible projects (i.e., offered on a drop-in basis) than well-defined and limited projects (i.e., set number of sessions offered). For example, administering the follow-up participant survey may not be feasible if participants do not return to attend more sessions of a flexible project.

## Supporting information

**S1 File. Evaluation map: Evaluation domains, questions and data sources.**
(DOCX)

**S2 File. Brief description of the 17 projects funded by the SWW program.**
(DOCX)

## Author contributions

**Conceptualization:** Vanessa Rose, Rhiannon Watt, Cara Büsst.

**Data curation:** Chloe Ang, Janell Kwok, Katherine Young.

**Investigation:** Thomas Steele, Chloe Ang, Vanessa Rose, Gayatri Kembhavi, Jamie Rowland.

**Methodology:** Vanessa Rose, Katherine Young, Sara Whittaker, Natasha Brusco.

**Project administration:** Thomas Steele, Chloe Ang.

**Supervision:** Vanessa Rose, Maryanna Abdo, Rhiannon Watt, Cara Büsst.

**Writing – original draft:** Thomas Steele, Chloe Ang, Vanessa Rose, Gayatri Kembhavi.

**Writing – review & editing:** Thomas Steele, Chloe Ang, Vanessa Rose, Gayatri Kembhavi, Jamie Rowland, Maryanna Abdo, Janell Kwok, Katherine Young, Sophie Merryfull, Sara Whittaker, Natasha Brusco, Rhiannon Watt, Cara Büsst.

## Acknowledgments

We would like to acknowledge the contributions of Anna Williamson, Chloe Jacob and Ivy Lim-Carter in the conceptualisation and early stages of designing this evaluation, as well as staff from each of the SWW funded project teams for their engagement with the evaluation.

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
