## [Decision Letter · Decision Letter 0]

2 Dec 2025

Protocol for evaluation of Movember’s Scaling What Works grant funding program- supporting the delivery of mental health interventions for men & boys in Australia, Canada and the United Kingdom

PONE-D-25-11964

Dear Dr. Thomas Steele

We’re pleased to inform you that your manuscript has been judged scientifically suitable for publication and will be formally accepted for publication once it meets all outstanding technical requirements.

**Comments from the editorial office** : Upon internal evaluation of the reviews provided, we kindly request you to disregard the reviewer report provided by Reviewer 1. No amendments are required in response to Reviewer 1’s comments. 

Kind regards,

Dirceu Henrique Paulo Mabunda, M.D.

Academic Editor

PLOS ONE

Additional Editor Comments (optional):

Address the reviewers comments and issues.

Reviewers' comments:

Reviewer's Responses to Questions

**Comments to the Author**

1. Does the manuscript provide a valid rationale for the proposed study, with clearly identified and justified research questions?

Reviewer #1: Yes

2. Is the protocol technically sound and planned in a manner that will lead to a meaningful outcome and allow testing the stated hypotheses?

Reviewer #1: Yes

3. Is the methodology feasible and described in sufficient detail to allow the work to be replicable?

Reviewer #1: Yes

4. Have the authors described where all data underlying the findings will be made available when the study is complete?

Reviewer #1: No

5. Is the manuscript presented in an intelligible fashion and written in standard English?

Reviewer #1: Yes

You may also provide optional suggestions and comments to authors that they might find helpful in planning their study.

Reviewer #1: The manuscript presents an important and well-designed protocol for evaluating Movember’s Scaling What Works (SWW) program across multiple community, school, and workplace interventions. The focus on scalability and cost-effectiveness within a real-world philanthropic funding context is highly innovative and of significant value to the field of men’s mental health research. The mixed-methods approach and the effort to capture implementation, economic, and outcome data across diverse projects are notable strengths.

At the same time, several areas of the manuscript would benefit from clarification or expansion to strengthen its methodological transparency and ensure the utility of the results for researchers, practitioners, and policymakers. The detailed comments are provided below for your consideration.

Major Comments

1. Project Heterogeneity and Statistical Analysis

• Issue: The diversity of funded projects (populations, settings, mechanisms) challenges comparability of pooled outcomes. While mixed-effects models are proposed, the analytic strategy lacks detail on how heterogeneity will be addressed.

• Location: Page 8 (“Evaluation Design”), Page 17 (“Analytical Approach”).

• Suggestion: Expand the analysis plan to clarify use of subgroup analyses, sensitivity testing, or stratification, and explicitly acknowledge the limits of cross-project pooling.

2. Scalability Assessment Tool Development and Validation

• Issue: The ISAT-inspired tool is innovative, but details on piloting, reliability, and validity assessment are limited.

• Location: Pages 12–13 (“Scalability Assessment”), Pages 20–21 (“Discussion”).

• Suggestion: Outline how inter-rater reliability, test–retest stability, or validity checks will be conducted, and describe plans for reporting these results.

3. Outcome Measures — Standardization and Psychometric Properties

• Issue: The Personal Wellbeing Index (PWI) is pragmatic but may lack sensitivity across contexts. Allowing projects to self-nominate additional outcomes introduces inconsistency, especially as not all nominated measures may have strong psychometric properties.

• Location: Page 15 (“Participant Survey Data”), Pages 23–24 (“Limitations”).

• Suggestion: Clarify minimum criteria for acceptable project-specific outcomes (e.g., validated instruments) and how inconsistent measures will be interpreted.

4. Handling of Missing Data and Attrition

• Issue: Missing data and high attrition are likely in flexible or drop-in projects, yet strategies for mitigation are not specified.

• Location: Page 17 (end of “Analytical Approach”), Page 24 (“Limitations”).

• Suggestion: Add detail on analytic methods for handling missingness (e.g., multiple imputation, sensitivity analysis), and plans to assess bias from attrition.

5. Economic Evaluation and Interpretation Beyond PWI

• Issue: Incremental cost per one-point PWI improvement is innovative, but unfamiliar to many stakeholders who expect QALY-based metrics. The manuscript acknowledges this but does not explore bridging interpretations.

• Location: Page 19 (“Cost-Effectiveness and Effect Analysis”), Page 24 (“Limitations”).

• Suggestion: Discuss how PWI-based results could be interpreted alongside more familiar economic measures, and clarify implications for policy translation.

6. Data Collection Support and Quality Assurance

• Issue: Heavy reliance on project teams for data collection risks inconsistent quality given varying evaluation capacity. While training and templates are noted, more robust QA procedures are not described.

• Location: Page 24 (“Limitations and Data Collection Challenges”).

• Suggestion: Provide details on data quality assurance (e.g., spot audits, automated data checks, cross-validation with administrative data).

7. Timeline and Milestone Visualization

• Issue: Timelines for recruitment, data collection, and reporting are dispersed across text, which makes it harder to follow.

• Location: Page 7 (“Methods/Design”).

• Suggestion: Include a schematic timeline (e.g., Gantt chart or figure) summarizing recruitment, follow-up, analysis, and reporting milestones.

8. Patient and Public Involvement (PPI)

• Issue: There is no reference to involvement of men/boys, participants, or broader stakeholders in study design, evaluation, or dissemination.

• Location: Not currently included (between “Evaluation Design” p. 8 and “Ethics Statement” p. 7–8).

• Suggestion: Add a section describing whether and how PPI was considered or planned.

Minor Comments

1. Generalizability

• Issue: The study is limited to three high-income countries.

• Location: Pages 3–5 (“Introduction”), Page 24 (“Limitations”).

• Suggestion: Explicitly acknowledge this and recommend replication in low- and middle-income contexts.

2. Timeline Clarity

• Issue: Data collection extends to 2026, but milestones are scattered.

• Location: Page 7 (“Methods/Design”).

• Suggestion: Reinforce with a single consolidated figure/timeline (ties with Major Comment #7).

3. Terminology

• Issue: Scaling, replication, and expansion are clearly distinguished but presented narratively.

• Location: Pages 4–5 (“Introduction”).

• Suggestion: Present these definitions in a concise table or schematic.

4. Data Availability Statement

• Issue: Data availability is promised post-study but without specifying repositories.

• Location: Pages ii–iii (“Data Availability Statement”).

• Suggestion: State intended repositories (e.g., OSF, institutional) to align with open science practices.

5. Editorial and Formatting Consistency

• Issue: Minor inconsistencies remain (e.g., acronym expansion for “PWI”; hyphenation in “knowledge base”).

• Location: Throughout; especially Abstract (Page 2) and first use in tables/figures.

• Suggestion: A thorough copyediting pass would resolve these.

**Do you want your identity to be public for this peer review?** For information about this choice, including consent withdrawal, please see our Privacy Policy

Reviewer #1: No

---

## [Editor Report · Acceptance letter]

PONE-D-25-11964

PLOS One

Dear Dr. Steele,

I'm pleased to inform you that your manuscript has been deemed suitable for publication in PLOS One. Congratulations! Your manuscript is now being handed over to our production team.

Kind regards,

on behalf of

Dr. Dirceu Henrique Paulo Mabunda

Academic Editor

PLOS One